# Genetic Diversity and Selection Signal Analysis of Hu Sheep Based on SNP50K BeadChip

**DOI:** 10.3390/ani14192784

**Published:** 2024-09-26

**Authors:** Keyan Ma, Juanjuan Song, Dengpan Li, Taotao Li, Youji Ma

**Affiliations:** College of Animal Science and Technology, Gansu Agricultural University, Lanzhou 730070, China; mky0017@163.com (K.M.); song25286@163.com (J.S.); ldp7208@163.com (D.L.); litt@gsau.edu.cn (T.L.)

**Keywords:** Hu sheep, SNP50K BeadChip, Genetic diversity, Selective sweep, ROH

## Abstract

**Simple Summary:**

This study focused on 50 Hu sheep from Gansu to analyze the genetic diversity and kinship relationships within the population and to construct family pedigrees. Candidate genes related to growth development and high reproductive performance in Hu sheep were identified. These findings provide a reference for future Hu sheep breeding, genetic evaluation, and population utilization.

**Abstract:**

This research is designed to examine the genetic diversity and kinship among Hu sheep, as well as to discover genes associated with crucial economic traits. A selection of 50 unrelated adult male Hu sheep underwent genotyping with the SNP50K BeadChip. Seven indicators of genetic diversity were assessed based on high-quality SNP data: effective population size (*Ne*), polymorphic information content (*PIC*), polymorphic marker ratio (*P_N_*), expected heterozygosity (*He*), observed heterozygosity (*Ho*), effective number of alleles, and minor allele frequency (*MAF*). Plink software was employed to compute the IBS genetic distance matrix and detect runs of homozygosity (ROHs), while the G matrix and principal component analysis were performed using GCTA software. Selective sweep analysis was carried out using ROH, Pi, and Tajima’s D methodologies. This study identified a total of 64,734 SNPs, of which 56,522 SNPs remained for downstream analysis after quality control. The population displayed relatively high genetic diversity. The 50 Hu sheep were ultimately grouped into 12 distinct families, with families 6, 8, and 10 having the highest numbers of individuals, each consisting of 6 sheep. Furthermore, a total of 294 ROHs were detected, with the majority having lengths between 1 and 5 Mb, and the inbreeding coefficient *F_ROH_* was 0.01. In addition, 41, 440, and 994 candidate genes were identified by ROH, Pi, and Tajima’s D methods, respectively, with 3 genes overlapping (*BMPR1B*, *KCNIP4,* and *FAM13A*). These results offer valuable insights for future Hu sheep breeding, genetic assessment, and population management.

## 1. Introduction

Hu sheep, originating from northern China, are recognized as excellent autochthonous breeds in China. They are characterized by their strong adaptability, good reproductive performance, rapid growth and development [1]. Currently, Hu sheep serve as excellent maternal resources for producing fat lambs, improving low-reproductive-efficiency sheep breeds. However, their widespread use has also compromised the purebred advantage of traditional breeding areas, leading to the degeneration of Hu sheep germplasm and confusion in pedigree records. Consequently, it is imperative to conduct genetic structure analysis and kinship identification of Hu sheep populations through various effective means to improve pedigree records at breeding farms and to provide a theoretical basis for the utilization, protection, and development of Hu sheep germplasm resources. 

Single nucleotide polymorphisms (SNPs) are third-generation molecular markers known for their wide distribution, genetic stability, and ease of detection [2]. The SNP chip, with advantages such as high density, wide coverage, and low cost, is one of the important breeding tools at present, and it has been widely used in pigs [3], cattle [4], sheep, [5,6], and goats [7]. The genetic diversity and structure of Kirghiz sheep were assessed using the SNP50K BeadChip by Li et al. [8], revealing abundant genetic diversity and minimal inbreeding, with the importance of enhancing selection in breeding rams underscored. Genetic diversity and inbreeding in Sumava and Wallachian sheep were evaluated by Karolína Machová et al. [9] using the Ovine SNP50K BeadChip, uncovering breed-specific genetic traits. A detailed analysis of genomic diversity and genes related to high-altitude adaptation in four goat populations—Tibetan goats, Taihang goats, Jining grey goats, and Meigu goats—was conducted using the SNP53K array by Zhong et al. [2], which provided substantial genomic evidence for future studies on the diversity of native Chinese goat breeds. Runs of homozygosity (ROHs) are defined as extended contiguous segments of homozygous genotypes inherited from a common ancestor [10]. The genome-wide inbreeding coefficient derived from ROHs (*F_ROH_*) provides a more accurate estimate of true inbreeding than the pedigree-based inbreeding coefficient [11]. Extended ROHs segments are indicative of recent inbreeding events, while shorter ROH segments suggest inbreeding that occurred in more distant generations [10]. Indeed, ROHs are also commonly used to identify genomic regions under putative selection [11].

Selective sweep analysis enables the identification of genomic regions experiencing selection pressures within a population [12]. This approach aids in pinpointing selected genes and clarifying the molecular mechanisms driving adaptation, evolution, and domestication at the genetic level [12,13]. Nucleotide diversity (Pi), which represents the genetic diversity within a population to some extent, can be assessed using π analysis [14]. This method calculates the average pairwise nucleotide sequence differences within a population and analyzes genetic diversity by comparing variations in π values [14]. Tajima’s D is another analytical approach for assessing genetic diversity within a population. Under neutral evolution, the mutation rate and genetic drift rate are expected to be balanced, resulting in a Tajima’s D value close to zero, indicating no genetic differentiation within the population [15]. However, under selective pressure, Tajima’s D values often become negative. By employing Tajima’s D analysis, one can identify genomic regions that deviate from neutrality, potentially highlighting target areas subject to selection [15].

By leveraging the Ovine SNP50K BeadChip, an initial assessment was made of the Hu sheep’s genetic diversity, population structure, kinship, and inbreeding levels, which then led to the demarcation of distinct family lines. Subsequently, the analysis of selection signals was conducted with the objective of pinpointing candidate genes linked to the growth and reproductive capabilities of Hu sheep. The insights garnered are intended to serve as a reference for the conservation, selective breeding, and innovative utilization of the Hu sheep genetic resources.

## 2. Materials and Methods

### 2.1. Sample Collection and Ethical Approval

In this study, the 50 Hu sheep used in the trial were selected from Lanzhou Xinyuan Modern Agriculture Technology Development Co., Ltd. (Lanzhou, China). They meet the characteristics and features of the breed, and each sheep has complete pedigree information. Blood samples of 5 mL were collected from the jugular vein of each ram using tubes containing EDTA as an anticoagulant, with each sample assigned a distinct identifier. The samples were then transported to the laboratory within 24 h and preserved at −80 °C for future DNA extraction. All experiments were conducted in accordance with the National Laboratory Animal Welfare instructions (2006-398) and were authorized by the Ethics Committee of the Laboratory Animal Center at Gansu Agricultural University (GSAU-Eth-ASF2022-008).

### 2.2. DNA Extraction, Genotyping and Quality Control

First, the CWE9600 Magbead Blood DNA Kit was used to extract DNA from each sample using magnetic bead technology. Subsequently, the concentration of genomic DNA was precisely measured using Qubit(Waltham, MA, USA), and the purity of DNA samples was assessed with Nanodrop (Thermo Fisher Scientific, Waltham, MA, USA) (OD260/OD280 = 1.7–2.1).Finally, the integrity of genomic DNA was evaluated using agarose gel electrophoresis to determine if there was any degradation. After performing quality assurance on the DNA, the samples were genotyped using the SNP50K array. Subsequently, SNP data underwent rigorous quality control via Plink v1.9 (Massachusetts General Hospital, Boston, MA, USA) [16] under the specified criteria: (1) minor allele frequency (MAF) > 0.05, (2) Hardy Weinberg equilibrium (HWE) > 10^−6^, and (3) SNP loci with a detection rate exceeding 90%, (4) excluding SNP loci located on sex chromosomes (parameters: –maf 0.05 –hwe 10^−6^ –geno 0.1 –chr 1–26).

### 2.3. Genetic Diversity 

#### 2.3.1. Effective Population Size (Ne) 

*Ne* denotes the theoretical population magnitude exhibiting identical genetic variance or an equivalent increase in inbreeding coefficient (rate of heterozygosity decline) as the actual population [17]. The methodologies proposed by Herrero-Medrano et al. [18] were utilized in the analysis, with the following formula used,
$$Ne=(1/4c)\times (1/r^{2}-1)$$
where *r*² denotes the LD between SNP loci, and *c* signifies the genetic distance between SNP loci measured in centimorgan (cM).

#### 2.3.2. Heterozygosity Analysis

Observed (*Ho*) and expected (*He*) heterozygosity are evaluated using the methodological framework proposed by Sun et al. [19]. The formula is
$$Ho=\frac{1}{N}\sum_{k=1}^{N}\frac{H_{k}}{n}\;\;\;\;,\;\;\;\;He=\frac{2_{n}}{2n-1}\frac{1}{N}\sum_{k=1}^{N}1-\sum p_{ki}^{2}$$
where *n* is defined as the total population size, *N* is the total number of loci, *H_k_* is indicated as the number of heterozygous individuals at locus k, and *P_ki_* is referred to as the frequency of allele i at locus k.

#### 2.3.3. Polymorphism Information Content (PIC) 

The *PIC* can be calculated using the formula proposed by Botstein [20],
$$PIC=1-\sum_{i-1}^{n}P_{i}^{2}-\sum_{i=1}^{n-1}\sum_{j=i+1}^{n}{2P}_{i}^{2}P_{j}^{2}$$
where *P_i_* and *P_j_* represent the frequencies of the ith and jth alleles, respectively, and *n* is the number of alleles.

#### 2.3.4. Polymorphic Marker Ratio (PN)

The minor allele frequency for each locus was calculated using Plink software [20], and *P_N_* was subsequently computed with a custom R script. *P_N_* was determined based on the following formula,
$$P_{N}=M/N$$
where *M* represents the number of polymorphic loci, and *N* denotes the total number of loci.

#### 2.3.5. Minor Allele Frequency (MAF)

MAF, the minor allele frequency, generally refers to the frequency of occurrence of uncommon alleles in a given population.

#### 2.3.6. Effective Number of Allele

The effective number of alleles refers to the number of alleles required to produce the same level of homozygosity observed in an actual population, assuming all alleles have equal frequencies in an ideal population. It is equal to the reciprocal of the observed homozygosity in the population.

### 2.4. Kinship Analysis

#### 2.4.1. Genetic Relationship Analysis Based on the G Matrix

The VanRaden’s GBLUP (Genomic Best Linear Unbiased Prediction) method [21] was employed to construct the G matrix. For this study, GCTA (V1.94) software (Westlake university, HangZhou, China) was applied to calculate the genetic relatedness coefficient between individuals, using the specific formula for computation
$$G=\frac{ZZ’}{2\sum P_{i}(1-P_{i})}$$
where (*P_i_*) represents the frequency of the ith allele.

#### 2.4.2. Genetic Distance Analysis

Genetic Distance (*D*) refers to the probability of non-homology between two individuals at the genomic level, thus defining *D* as the genetic distance between two individuals.
$$D=1-D_{ST}$$

*D_ST_* refers to the probability of homology between two individuals at the genomic level. The formula for *D_ST_* calculation is as follows:$$D_{ST}=\frac{0.5*IBS1+IBS2}{N}$$

*IBS*1 refers to the number of pairs of loci where one observed value is the same, while *IBS*2 denotes the number of loci pairs where both observed values are identical. *N* represents the total number of marker loci. Both *IBS*1 and *IBS*2 are computed using Plink (V1.90) software [20]. 

Heatmaps of the G matrix and IBS distance matrix are generated using R software (V3.5.3) (Mathsoft, Nattick, MA, USA) (pheatmap package) [10].

### 2.5. Population Structure Analysis 

#### 2.5.1. Principal Component Analysis

Based on the filtered SNP data, PCA analysis was conducted using GCTA software (V1.94) [22] to calculate the first two principal components and determine the clustering of individual Hu sheep.

#### 2.5.2. Cluster Analysis and Pedigree Construction

Samples are organized into clusters based on the genetic distance matrix derived from genetic distance analysis using the neighbor-joining (NJ) method. Individuals with close relationships are placed into smaller groups, while those with more distant relationships are categorized into larger ones, continuing until all samples are included. The relationships among all samples in terms of proximity and separation are highlighted through this method. After combining the results of genetic relationship analysis and cluster analysis according to the method [23], clustering is performed using the neighbor-joining method [24], with the criterion that the molecular genetic relationship between rams is greater than or equal to 0.1, and different pedigrees are divided.

### 2.6. Run of Homozygosity Detection

According to the method proposed by Signer-Hasler et al. [25], PLINK software [20] was used for whole-genome ROH detection in Hu sheep. The analysis utilized the following criteria: (1) a sliding window consisting of 50 SNP sites, (2) ensuring that the overlap proportion of homozygous segments within each window exceeded 0.05, (3) each ROH segment contained a minimum of 100 consecutive SNPs, (4) a SNP marker density of at least 50 kb per SNP, (5) limiting the distance between adjacent SNPs within homozygous segments to less than 100 kb, and (6) permitting a maximum of two missing SNPs and one heterozygous site per ROH. The genomic inbreeding coefficient *F_ROH_* is calculated based on the detected ROH:$$F_{ROH}=\frac{\sum L_{ROH}}{L_{auto}}$$
where $\sum L_{ROH}$ represents the sum of the lengths of *ROH* segments on autosomes, and $L_{auto}$ represents the total physical length of autosomes.

### 2.7. Selective Sweep Analysis

To identify the genomic regions frequently associated with ROHs, the percentage of SNP occurrences within ROHs was computed by recording how often each SNP was detected across individuals. The most prevalent genomic regions linked to ROHs were then determined by selecting the top 1% of SNPs observed within these ROHs (Appendix A). Adjacent SNPs exceeding this threshold were consolidated into genomic regions referred to as ROH islands [10,11]. In addition, VCFtools software (v0.1.16) [26] was used to identify genomic regions under selection by setting a window size of 100,000 bp and a step size of 10,000 bp. Pi and Tajima’s D were employed respectively for this purpose. Gene annotation for these selection signal regions was conducted using the reference genome Oar_v4.0. DAVID [27] was also used to select the GO and KEGG pathways that were significantly enriched in candidate genes. The biological functions of the genes within these islands were investigated through a thorough review of the relevant literature.

## 3. Results

### 3.1. Summary of Data

Following rigorous quality control procedures, 56,522 high-quality SNPs were retained for subsequent analyses (Table 1). The quantity of SNPs before and after quality control is illustrated in Appendix A. 

### 3.2. Genetic Diversity

To assess the genetic diversity of the Hu sheep population, this study examined six genetic diversity parameters using SNP data. As presented in Table 2, the Hu sheep demonstrated an average *Ne* of 6.9, *P_N_* of 0.873, *He* of 0.356, *Ho* of 0.36, *PIC* of 0.264, effective number of alleles of 1.563, and *MAF* of 0.253. SNPs with *PIC* values ranging from 0 to 0.15 constituted only 18.55%, while those within the ranges of 0.15–0.30 and 0.30–0.45 accounted for 28.03% and 53.42%, respectively (Figure 1A). The Ho for all loci in the population ranges from a maximum of 0.826 to a minimum of 0, while the He ranges from a maximum of 0.5 to a minimum of 0.0198. *Ho* (0.36) closely mirrored *He* (0.356) in this population. The effective number of alleles ranges from a maximum of 1.734 to a minimum of 1.523. Additionally, the *MAF* distribution was relatively homogeneous, with the lowest proportion observed in the 0.1–0.2 range at 17.34% and the highest in the 0–0.1 range at 21.45% (Figure 1B).

### 3.3. Kinship Analysis

Drawing upon the G matrix analysis (Appendix A), a deeper shade of purple signifies a stronger genetic kinship among Hu sheep individuals. As Figure 2A illustrates, the majority of the sheep population demonstrates remote genetic connections, albeit with a handful showing a closer familial bond. Additionally, the IBS genetic distance assessment disclosed distances varying between 0.222 and 0.294 across the Hu sheep cohort, averaging 0.286 (Appendix A). The IBS matrix heatmap concurs with the G matrix findings, jointly highlighting that most Hu sheep maintain distant genetic ties (Figure 2B).

### 3.4. Population Structure Analysis

The PCA result, visualized in Figure 3A, reveals that the first and second principal components contribute to explaining 62.1% and 24.1% of the variation, respectively. Despite the majority of the Hu sheep population assembling in a cohesive group, there remains a select few individuals (17, 47, 39) that manifest significant genetic disparities from their counterparts. As depicted in the phylogenetic tree in Figure 3B, the 50 individual Hu sheep predominantly aggregate into three principal clusters, further distinguished into 12 distinct families according to their genetic affiliations. Of these families, families 6, 8, and 10 are distinguished by hosting the greatest number of rams, encompassing a total of six individuals, whereas family 2 has the fewest rams, with only two individuals.

### 3.5. ROH Analysis 

Within the Hu sheep population, a total of 294 ROHs were identified. Out of these, 77.55% fall within the 1–5 Mb length bracket, 15.99% occupy the 5–10 Mb range, and merely 6.46% surpass 10 Mb in length (Table 3). The briefest ROH, roughly 1 Mb long, resides on chromosome 13; conversely, the most extended ROH, stretching across 26 Mb, is situated on chromosome 3. Shifting to an individual viewpoint (Figure 4A), the peak count of ROHs discovered in a single Hu sheep amounts to 20, whereas the minimum tally is 1. Specifically, among 45 sheep, the cumulative length of ROHs remains under 50 Mb. However, there’s one notable exception where this figure surpasses 100 Mb. Furthermore, four sheep exhibit a total ROH span ranging from 50 to 100 Mb. When it comes to chromosomal distribution, ROHs are not uniformly spread. Chromosome 26 boasts just a solitary ROH, whereas chromosome 6 leads the pack with a notable 43 ROHs. In terms of ROH coverage, chromosome 2 takes the crown with a rate of 65.57%, whereas chromosome 25 lags behind with a mere 9.69% coverage (Figure 4B). The inbreeding coefficient F_ROH_ based on ROHs in the Hu sheep population is 0.01 (Figure 4C).

### 3.6. Candidate Gene Identification and Enrichment Analysis

The frequency of SNPs observed across all individuals was calculated to assess the proportion of SNPs in ROHs. We detected ROH islands as genomic regions that showcased the highest 1% frequency of SNPs associated with ROHs (Appendix A). After merging ROHs, a total of 12 ROH islands were identified, which were subsequently annotated, yielding 41 candidate genes, such as *PYURF*, *BMPR1B*, *PPARGC1A*, *PDHA2*, *LAP3*, and *PDLIM5* (Appendix A). A total of 624 selective regions were identified based on Pi (Figure 5A), which were annotated to yield 440 candidate genes, including *FGFR1*, *PTPRR*, *SOS2*, *BMPR1A*, *ZBP2*, and *TYRO3* (Appendix A). Based on Tajima’s D, a total of 2561 selection regions were detected (Figure 5B). After annotation, 994 candidate genes were identified, including *BMPR1B*, *RPGRIP1*, *KCNMB2*, and *ACTG1* (Appendix A). Notably, three genes were detected by ROH, Pi, and Tajima’s D such as *BMPR1B*, *KCNIP4*, and *FAM13A* (Figure 5C). Carrying out detailed functional enrichment analysis on these nominated genes disclosed their substantial participation in various biological processes, including the positive regulation of muscle tissue development, animal organ development, spermatogenesis, and the binding of sperm to zona pellucida. Additionally, these genes are also found to be crucial in several vital pathways, such as the citrate cycle, pyruvate metabolism, and the TGF-beta signaling pathway (Figure 6A–F).

## 4. Discussion

Livestock genetic resources play a vital role in biodiversity, possessing unique characteristics that cannot be replaced [10,11]. The variety within these resources provides the basis for breeding superior livestock breeds capable of adapting to unforeseen demands. Therefore, it is crucial to conserve and utilize these resources to enhance the overall strength of the industry, its capacity for sustainable development, genetic source self-sufficiency, and market competitiveness. Hu sheep are celebrated for their rapid early growth, high fertility, and ability to thrive in hot and humid conditions [1]. However, recent market pressures have resulted in unplanned crossbreeding with numerous high-quality meat-producing sheep breeds like Suffolk and Dorper to address deficiencies in meat production among Hu sheep [28]. This has led to a significant decline in purebred Hu sheep populations, resulting in the loss of their exceptional genetic traits and posing a severe risk of extinction for the breed.

Thus, it is crucial to scientifically evaluate the conservation impact on Hu sheep, as these findings could provide valuable insights for the sustainable progression of subsequent conservation efforts. In our study, we calculated seven genetic diversity parameters to assess the genetic variability within the Hu sheep population. Effective population size (*Ne*), for instance, denotes the ideal population size where gene frequency variance or heterozygosity decay matches that of the actual population. A smaller *Ne* indicates decreased genetic variability within the population, potentially hindering future development [29]. Our results revealed an *Ne* of 6.9 for Hu sheep, which is higher than that observed in Kirgiz sheep (1.583) [8], Tezanghan sheep (1.59) [30], and Jining gray goats (1.697) [31]. Polymorphic information content (*PIC*) measures the frequency and number of alleles in a population, reflecting the degree of polymorphism at a locus and assessing genetic diversity [11]. In our study, the *PIC* value for Hu sheep was 0.264, which is close to that of Minxian black fur sheep (0.273) and Tan sheep (0.275) [11], and Boer goat (0.271) [10], indicating Hu sheep have a moderate level of genetic variation and relatively high genetic diversity. The polymorphic marker ratio (*P_N_*) indicates the proportion of polymorphic loci, with higher values indicating richer genetic information. Our findings revealed a *P_N_* value of 0.873 for Hu sheep, which is close to that of Kirgiz sheep (0.889) [8]. Heterozygosity serves as another parameter to gauge genetic diversity within a population. Higher heterozygosity within a breed signifies greater genetic diversity [32]. In our study, the observed heterozygosity of Hu sheep was 0.36, which is higher than that of Yongdeng Qishan sheep (0.199) [11], Lanzhou fat-tailed sheep (0.341) [11], and Hainan black goat (0.2) [33], and close to the expected heterozygosity (0.356). This indicates that the Hu sheep population has high genetic diversity and a low level of inbreeding within the population. Here, the *MAF* recorded for Hu sheep was 0.253. Research has indicated that *MAF* is crucial in assessing the biological importance of genetic diversity. Rare low-frequency mutations (i.e., those with lower *MAF*) might exert more potent functional impacts compared to frequent mutations (variations with higher *MAF*) [9]. Based on the research results presented above, Hu sheep possess high genetic diversity. 

Lineage plays a crucial role in animal genetics and breeding research. Historically, livestock and poultry identification relied heavily on pedigree records. However, issues like ear tag losses and recording errors in farm management can introduce inaccuracies into traditional pedigrees, resulting in incomplete or erroneous lineage records. Such discrepancies pose challenges in establishing robust breeding stocks or introducing new sheep breeds, thereby hindering effective breeding strategies. Studies indicate that pedigrees constructed using SNP BeadChip offer greater accuracy by leveraging genetic relationships and distances between individuals, surpassing the precision of traditional pedigree records [23]. Therefore, this research initially conducted kinship analysis using high-quality SNPs. Both the G matrix and IBS genetic distance matrix serve as effective tools for assessing individual kinship within populations. The G matrix provides a more precise depiction of genuine kinship connections among individuals compared to pedigree-based calculations, aligning closely with real-world scenarios [11]. IBS focuses solely on the similarity of genetic markers or alleles between individuals, enabling kinship analysis within populations even without clear pedigree records or ancestral samples [10]. The results from the G matrix and IBS genetic distance matrix exhibit strong consistency, with IBS values ranging from 0.222 to 0.294 and averaging 0.286. This figure is lower than that observed in Kirgiz sheep (0.294) [8] and Jining gray goats (0.33) [34], indicating distant kinship relationships predominate among most individuals in the Hu sheep population, with only a small minority showing closer kinship. These findings suggest a mild level of inbreeding within the population.

To delve deeper into the genetic structure of this group, PCA was conducted on a cohort of 50 Hu sheep. The results reveal a general clustering of all individuals with minimal genetic variance, though a few individuals fall beyond the confidence interval, hinting at potential gene flow with other sheep breeds. Furthermore, based on the analysis results from the G matrix and IBS matrix, we conducted clustering analysis and pedigree division. This was done using the criterion of kinship equal to or greater than 0.1 among Hu sheep individuals, following the methodology outlined by Li et al. [23]. The findings revealed that the phylogenetic tree displayed three major branches with the 50 Hu sheep. Subsequently, these 50 Hu sheep were classified into 12 distinct lineages. Notably, most lineages had few rams, highlighting the importance of careful breeding practices to avoid genetic diversity loss in future generations.

ROH are continuous segments of homozygous genotypes found within individual animals [10]. They arise when identical haplotypes are passed down from parents to offspring, providing insights into population history based on their length and frequency [35]. Longer ROH segments suggest recent inbreeding events, whereas shorter segments indicate more distant relationships [36]. In the present study, 294 ROHs were identified within the Hu sheep population. The majority of these ROHs were categorized as short, with lengths under 5 Mb, whereas ROHs exceeding 5 Mb represented only 22.45%. This indicates that the population has not experienced substantial inbreeding recently. ROHs are unevenly distributed across chromosomes; chromosome 6 exhibited the highest count of ROHs, totaling 43, whereas chromosome 26 showed the lowest number. Traditional methods for calculating inbreeding coefficients are based on pedigree data, which relies on the precision and completeness of the pedigree information available [34]. With the progress in sequencing technologies, new methods have been developed to compute inbreeding coefficients directly from whole-genome SNP data. Specifically, *F_ROH_* provides a precise measure of the percentage of homozygosity in individual genomes without needing pedigree data [37,38]. In this analysis, the *F_ROH_* for the Hu sheep population was found to be 0.01, which is significantly lower compared to other similar studies, such as those conducted on the Yongdeng Qishan sheep (0.1546) [11] and the large-tailed Han sheep (0.08) [39]. This also suggests that the level of inbreeding within the group is relatively low.

This study conducted a selective sweep analysis of Hu sheep using three methods, ROH, Pi, and Tajima’s D, with the aim of identifying candidate genes associated with economically important traits. The findings will provide a basis for molecular-assisted selection breeding in the near future. The ROH, Pi, and Tajima’s D approaches identified 41, 440, and 994 potential genes, respectively, with 3 overlapping genes: *BMPR1B*, *KCNIP4*, and *FAM13A*. Analysis of these candidate genes revealed that they are mainly involved in biological processes such as growth and development, spermatogenesis, and energy metabolism in Hu sheep. *PYURF* was notably enriched in biosynthetic and metabolic processes. Studies suggest that *PYURF* is significantly related to immunity and disease resistance in dairy cattle [40]. This finding was subsequently validated by Seyed Mohammad Ghoreishifar in his research on Swedish cattle breeds [41] and by Jelena Ramljak in her studies on Croatian sheep [42]. *BMPR1A* and *BMPR1B* are members of the BMPR family and are critical for cellular signaling and tissue development. In this study, they show significant enrichment in spermatogenesis and the TGF-beta signaling pathway. According to Fang et al. [43], *BMPR1A* and *BMPR1B*, as *BMP4* receptors, exhibit high expression levels in prepubertal testicular progenitor Leydig cells and isolated cells in mice, playing crucial roles in spermatogenesis. Additionally, another study highlighted that an increase in BMP4 expression in sheep testes leads to elevated levels of *BMPR1A* and *BMPR1B*, which, in turn, activate the SMAD1/5/8 signaling pathway to regulate spermatogenesis [44]. Moreover, *BMPR1B* is notably recognized as a key gene related to prolificacy in sheep; the *Fec*B variant of *BMPR1B* has a significant effect on ovulation rates and litter sizes in sheep [45]. *PDHA2* is notably prevalent in both the citrate cycle and the glycolysis/gluconeogenesis pathways. The citrate cycle is a fundamental process in cellular respiration, crucial for energy metabolism and for providing precursor molecules for biosynthetic pathways [46]. In the context of spermatogenesis, the citrate cycle, alongside glycolysis and gluconeogenesis, delivers critical energy and metabolic intermediates to sperm cells. This support facilitates the cells’ proliferation and differentiation, which, in turn, influences spermatogenesis. Studies have identified *PDHA2* as a potential new gene associated with male infertility. Its protein product is a mitochondrial enzyme that shows peak expression in sperm that has been ejaculated [47]. Research by Vivek Kumar has shown that *PDHA2*, a component of the pyruvate dehydrogenase complex, undergoes tyrosine phosphorylation during the capacitation phase of hamster sperm, indicating a potential role of *PDHA2* in sperm capacitation [48]. *ZPBP2*, found at the interface where sperm attaches to the zona pellucida, serves as a vital receptor for the zona pellucida on sperm and is crucial in sperm-egg interactions [49]. Additionally, the *CABS1* gene, which plays a role in spermatogenesis, is reported to be critical for the correct formation and movement of the sperm tail [50]. Chen et al. used CLR, π ratio, FST, and XP-EHH to detect the candidate genes characteristic of positive selection in Xilin buffalo and found that KCNIP4 plays a crucial role in reproductive regulation [51]. Additionally, studies have shown that KCNIP4 was found in highly differentiated regions of the Lingxian white goose genome, affecting its reproduction [52]. In a study comparing the proteomic characteristics of spermatogenic cells between hybrid cattle and indigenous cattle, it was found that the expression level of *FAM13A* was significantly up-regulated in the spermatogenic cells of indigenous cattle [53]. Overall, these candidate genes are involved in testicular development and spermatogenesis in various animals, which might shed light on the high reproductive efficiency observed in Hu sheep. They could serve as potential candidate genes for future molecular breeding.

*PPARGC1A* is prominently featured in several critical pathways, such as the Adipocytokine signaling pathway and the Insulin signaling pathway. This gene plays a role in numerous biological processes. For example, research has shown that reducing *PROX1* levels can boost the production of milk fatty acids in dairy goats through its impact on *PPARGC1A* [54]. Additionally, *PPARGC1A* facilitates mitochondrial biogenesis and influences skeletal muscle metabolism by regulating glycolysis and the TCA cycle, which, in turn, promotes growth and development [55]. Overexpression of *PPARGC1A* has been observed to mitigate the abnormal expression of genes related to steroidogenesis and testosterone production induced by miR-1197-3p, positioning it as a key regulator of Leydig cell testosterone secretion in the testes of goats [56]. In the glutathione metabolism pathway, both *HPGDS* and *LAP3* are notably enriched. The expression levels of *LAP3* affect the development of myocytes in sheep and may influence muscle development in sheep [57]. Meanwhile, Cai et al. have suggested that *LAP3* could be a candidate gene for growth and development in Wenchang chickens [58]. Conversely, *HPGDS* has been identified as having a significant role in the growth and development of yaks, making it a valuable candidate for marker-assisted selection (MAS) in yaks [59]. The *PDLIM5*, which is enriched in the process of muscle structure development, is a cytoskeleton-associated protein essential for various tissues and cells. This gene has been connected with muscle cell differentiation in Boer goats [60] and muscle development in Huaxi cattle [61]. *FGFR1* and *ALOX5* are associated with the pathways of positive regulation of cell population proliferation and arachidonic acid metabolism, respectively. Research indicates that *FGFR1* enhances the proliferation of myoblasts without influencing their differentiation, and its transcriptional activity is modulated by core promoter methylation levels [62]. Conversely, research by Yin et al. [63] has demonstrated that disrupting the *ALOX5* impedes both the proliferation and differentiation of myoblasts, whereas increasing *ALOX5* expression fosters both processes. These insights provide valuable theoretical knowledge regarding the mechanisms regulating muscle growth and development in Haiyang Yellow chickens. In summary, the above candidate genes are all involved in the regulation of animal growth and development and may serve as important references for improving animal production performance.

## 5. Conclusions

The SNP50K BeadChip was used in this study to analyze the genetic diversity and kinship relationships of 50 Hu sheep. These sheep were further categorized into families, and selective sweep analysis was conducted using three different methods. It was revealed that the Hu sheep population exhibits high genetic diversity. The 50 Hu sheep were classified into 12 families, and attention should be paid to the selection and breeding of offspring in the later stages to avoid the loss of bloodlines. Additionally, a series of candidate genes associated with growth, development, and high productivity were identified. These candidate genes are expected to be useful for future molecular-assisted selection breeding. This research provides a foundation that can be utilized for future breeding, genetic evaluation, and population management of Hu sheep.

## Figures and Tables

**Figure 1 animals-14-02784-f001:**
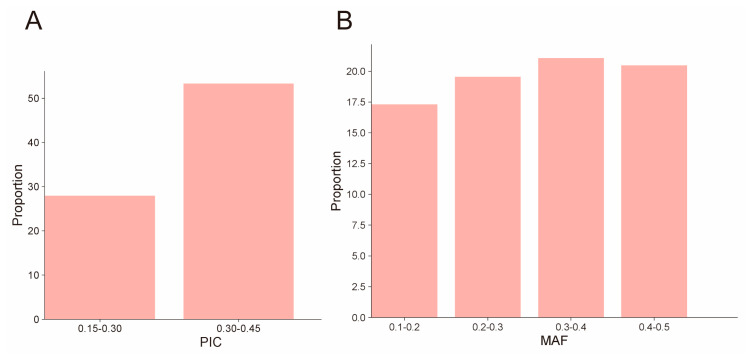
Genetic diversity analysis. (**A**) PIC distribution; (**B**) MAF distribution.

**Figure 2 animals-14-02784-f002:**
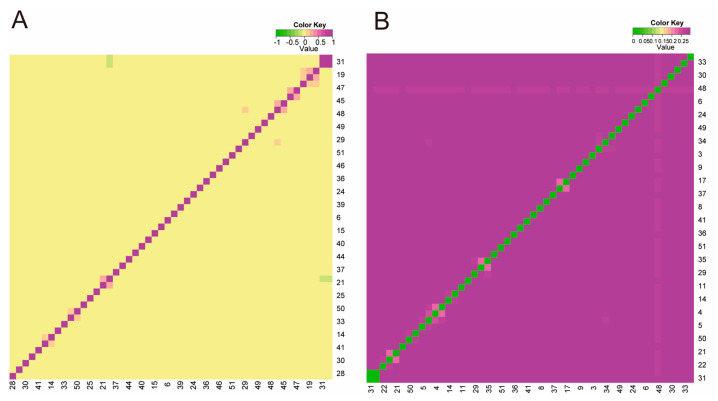
Heatmap of kinship of Hu sheep. (**A**) G-matrix heatmap (the closer the color is to purple, the closer the kinship); (**B**) IBS genetic distance matrix heatmap (the closer the color is to green, the closer the kinship).

**Figure 3 animals-14-02784-f003:**
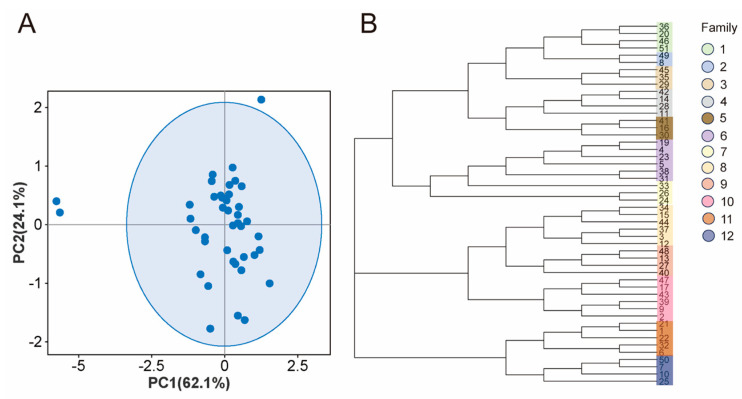
Analysis of population structure. (**A**) Principal component analysis; (**B**) phylogenetic tree.

**Figure 4 animals-14-02784-f004:**
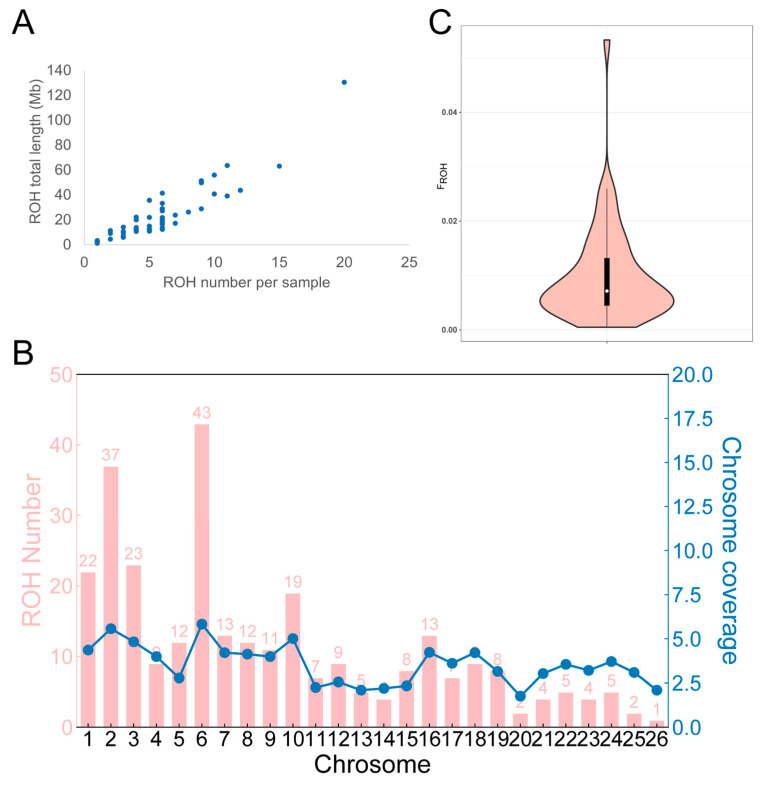
ROH statistics and *F_ROH_*. (**A**) Number and length statistics of ROH in each Hu sheep; (**B**) number of ROH per chromosome (bars) and average percentage of each chromosome covered by ROH; (**C**) *F_ROH_* of the Hu sheep population.

**Figure 5 animals-14-02784-f005:**
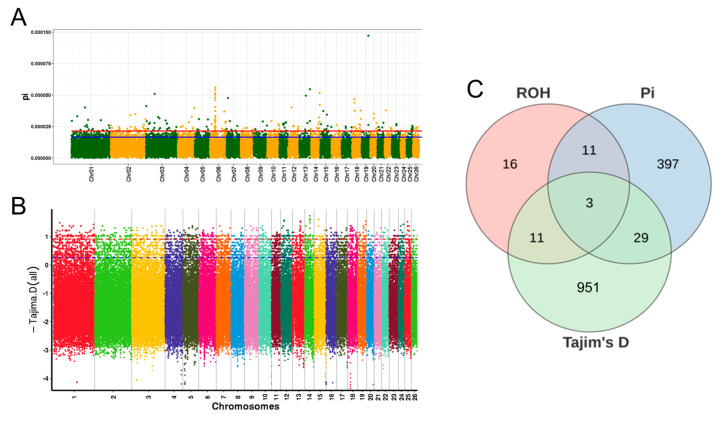
Detection of selected regions in the Hu sheep population. (**A**) Manhattan plot of selection signatures detected by Pi; (**B**) Manhattan plot of selection signatures detected by Tajim’s D; (**C**) number of candidate genes for ROH, Pi, and Tajima’s D.

**Figure 6 animals-14-02784-f006:**
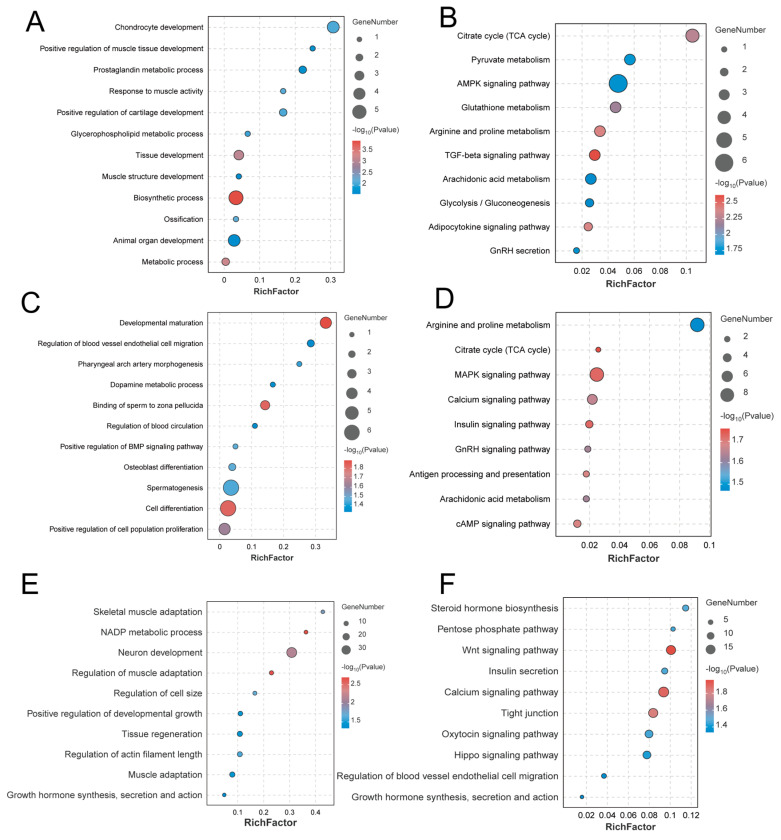
(**A**,**B**) Enrichment analysis results based on ROH (left for GO, right for KEGG); (**C**,**D**) Enrichment analysis results based on Pi (left for GO, right for KEGG); (**E**,**F**) Enrichment analysis results based on Tajima’s D (left for GO, right for KEGG).

**Table 1 animals-14-02784-t001:** Results of SNP quality control statistics.

Quality Control Standard	Number of SNPs
Total number of SNPs	64,734
SNP with MAF < 0.01	4123
SNP with P < 10^−6^ of Hardy-Weinberg equilibrium	105
SNP with call rate < 0.90	1197
SNPs on chromosome X	1527
SNPs on chromosome Y	1251
Insertion/Deletion	9
SNPs used after quality control	56,522

**Table 2 animals-14-02784-t002:** Results of genetic diversity analysis.

Index	Average Value
Effective population size (*Ne*)	6.9
Polymorphic information content (*PIC*)	0.264
Polymorphic marker ratio (*P_N_*)	0.873
Expected heterozygosity (*He*)	0.356
Observed heterozygosity (*Ho*)	0.36
Effective number of alleles	1.563
Minor allele frequency (*MAF*)	0.253

**Table 3 animals-14-02784-t003:** The statistical distribution of different lengths of ROH in Hu sheep.

ROH Length (Mb)	ROH number	Percent (%)
1–5	228	77.55
5–10	47	15.99
10–15	11	3.74
15–20	6	2.04
>20	2	0.68
Total	294	100

## Data Availability

The data presented in this study are available on request from the first author/corresponding authors.

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
