# Peer review of "Genetic Diversity and Selection Signal Analysis of Hu Sheep Based on SNP50K BeadChip"

_animals, 2024, doi:10.3390/ani14192784_

Round 1
Reviewer 1 Report (New Reviewer)
Comments and Suggestions for Authors
The work is a good attempt at investigating the genetic diversity of Hu sheep and kinship relationships within the population. In the research, Runs of Homozygosity (ROH) and the nucleotide diversity index (Pi) were employed to pinpoint candidate genes linked to Hu sheep adaptability and reproductive capabilities. The results showed a total of 294 ROHs and several candidate genes playing significant roles in physiological processes, such as spermatogenesis and muscle tissue development.
The work is ACCEPTABLE. There are no comments against the publication of the article. However, some specific and few comments are addressed to the authors.
Specific Comments
In my view, the work had already undergone a review, and a version with the corrections suggested by another reviewer was sent to me. Here are my comments:
1 – In line 126, this sentence can be removed.
2 – In lines 153-154, Plink software was not cited correctly. It is not included in the references.
3 - Lines 561 to 566 convey the same idea as lines 602 to 606. One of the two statements can be removed."
4 – In line 718, the authors Y et al. are not cited in the references.
5 - Draw a parallel between the sentences in lines 610-612 and lines 739-741 and check if these statements are not contradictory."
Author Response
Dear Reviewer, please find my response letter attached. Thank you for your hard work

Reviewer 2 Report (New Reviewer)
Comments and Suggestions for Authors
Dear Authors
Thank you for submitting your manuscript, “Genetic Diversity and Selection Signal Analysis of Hu Sheep Based on SNP50K BeadChip.” The goal of the study, which is to examine the genetic diversity and kinship among Hu sheep and to identify genes associated with crucial economic traits, is indeed of significant interest. However, after carefully reviewing your work, I regret to inform you that I cannot recommend it for publication in its current form.
The primary reasons for my decision are related to the quality of the scientific writing and the execution of the study. I encourage you to address the issues highlighted below and consider resubmitting the paper after substantial revisions.
**Key points that led to my decision:**
1. **Insufficient Literature Review:**
- The introduction lacks sufficient references and does not provide an in-depth literature review that adequately supports the study’s objectives and the significance of the breed under investigation.
2. **Incomplete Methodological Details:**
- The Materials and Methods section is lacking crucial details, making it difficult to assess the validity of your analyses. For example:
- How did you measure the effective population size?
- What tools or databases did you use for functional annotation and enrichment analysis?
- The genetic metrics mentioned are underexplained and need further elaboration (currently, only a few lines are provided).
3. **Methodological Concerns:**
- Some results raise questions about the methodologies employed. For instance, the reported high coverage of chromosomes by ROH (Runs of Homozygosity) seems inconsistent with the low levels of inbreeding you found. Without a clear explanation of how the effective population size was estimated, these results are difficult to reconcile.
4. **Discussion Section:**
- The discussion primarily compares your findings with other studies without thoroughly analyzing the implications for the Hu sheep breed. Additionally, concepts introduced in the discussion should be more prominently featured in the introduction to better frame the study.
5. **Conclusion Section:**
- The conclusion reads more like a summary of results rather than a synthesis of the findings and their broader significance.
**Recommendations for Improvement:**
1. Consider using the Tajima’s D test instead of Pi metrics. While Pi metrics provide the average genetic differences among individuals, Tajima’s D could offer insights into sites under selection over time.
2. Avoid mentioning analyses or approaches that were not actually performed. For example, the abstract suggests an association with SNP markers and traits, implying a GWAS analysis, which was not conducted. Similarly, in the discussion, the term “whole genome analysis” is misleading since whole-genome sequencing was not used.
3. Reflect on whether a sample size of just 50 animals is sufficient to represent the population effectively.
4. Please ensure that the manuscript submitted is the final version. The current document appears to have tracked changes still visible, which should be resolved before submission.
I hope these suggestions are helpful,
Best regards,
Author Response
Dear Reviewer, please find my response letter attached. In addition, I would like to explain that the manuscript has already undergone one round of peer review, and according to the editor’s requirements from Animals, all modifications must be highlighted using tracking mode. This may have caused some confusion. I apologize again for any issues this may have caused. Thank you for your understanding.
Thank you for your hard work.

Reviewer 3 Report (New Reviewer)
Comments and Suggestions for Authors
Revision to Manuscript Genetic Diversity and Selection Signal Analysis of Hu Sheep Based on SNP50K BeadChip (animals-3162073)
Mayor comment.
The authors mention that “a total of 50 adult rams without any genetic relationship….” Line 137.
Any additional information about the rams is provides in order to understand if the sample is representative, or which are the selection criteria.
During the quality control (section 2.2) using plink authors could confirm that all animals are unrelated. Plink can assess the genomic relationship.
In section 2.4, specifically in 2.4.1. Authors just mention the VanRaden matrix. In this case the selection of the kinship matrix is not irrelevant. The authors must explain further details or contrasting with others kinship matrices.
Please review the quality control, specially because the authors mention that “the IBS genetic distance assessment disclosed distances varying between 0.222 and 0.294 across the Hu sheep cohort, averaging at 0.286” (line 295-296).
I would suggest answer this comments and after continue with the revision.
Author Response
Dear Reviewer, please find my response letter attached. Thank you for your hard work.

Round 2
Reviewer 2 Report (New Reviewer)
Comments and Suggestions for Authors
Thank you for addressing the points raised in the initial review. I appreciate the effort made to improve the manuscript, and I recognize the potential significance of the results presented. However, I still have several concerns regarding the interpretations of the data and the lack of appropriate citations throughout the manuscript. For these reasons, I cannot recommend the manuscript for publication without further major revisions. Below are specific points that still need to be addressed:
1. L21-22: "The analysis revealed that 55,522 SNPs were identified among the 50 Hu sheep." – This statement is misleading. The analysis you performed was limited to quality control, ensuring data consistency. The way it is phrased suggests that all 55,522 markers are linked to the genetic diversity of the animals, which is not accurate.
2. L22-23: The statement about low genetic diversity alongside the claim that most individuals are unrelated contradicts established literature on genetic diversity. Decades of research suggest otherwise. Please clarify or provide more justification.
3. L56-67: This section appears somewhat disconnected. While it’s important to include prior studies, you need to link this information more directly to the objectives of your current study.
4. L69-70: This sentence needs an appropriate citation to support the claim.
5. L70-72: Please provide a reference to substantiate this statement.
6. L76-77: A citation is required here.
7. L79-80: A citation is needed for this sentence.
8. L82-85: Please include a reference.
9. L91-92: Redundant definitions of ROH and Pi – these were already explained earlier in the manuscript and do not need to be defined again.
10. Quality Control for MAF and HWE: Why did you apply quality control for Minor Allele Frequency (MAF) and Hardy-Weinberg Equilibrium (HWE)? Fixed markers, often excluded during such quality control, can be informative in genetic diversity studies and provide insight into significant evolutionary processes within populations.
11. L159: The work by VanRadem should be cited, and you should specify which of the three methods proposed by VanRadem you used for the G matrix.
12. L166: The sentence should start as "Genetic Distance (D)".
13. L175: A reference for the PLINK software is required.
14. L177: A reference for the R software should be provided.
15. L181: Please include a reference for the GCTA software.
16. L191: Specify which method was used in this section.
17. L196: A reference for the PLINK software should be provided here as well.
18. L215: Please include a reference for the DAVID tool.
19. Additionally, you performed gene annotation and functional analysis for Pi and Tajima’s D. Please correct the information in the text to reflect this.
20. L222: Similar to point 1, the statement here is misleading and should be revised.
21. L224-227: This is not your result; I suggest you consider removing this section.
22. Regarding the effective population size: To which generation does the value of Ne correspond? Did you apply any quality control measures for Linkage Disequilibrium (LD) distances, such as requiring a minimum number of SNP pairs in LD to make the estimate reliable? How many SNP pairs in LD were available to estimate the current generation?
23. L246: You didn’t use the whole genome in your analysis; this should be clarified.
24. For the Principal Component Analysis (PCA), did you perform pruning for LD? This is important to ensure marker independence in the analysis.
25. L331-334: A reference is needed to support this statement.
26. L336: Why are you comparing your results to pig populations? This comparison requires further explanation.
27. L357-369: There’s a contradiction here: you claim minimal inbreeding but recommend controlling inbreeding in future breeding programs. You also state that genetic diversity is limited, but this conflicts with the claim of minimal inbreeding. This needs clearer explanation. For example, you could clarify that “If the results show minimal inbreeding, it suggests that the population has not yet experienced significant inbreeding effects. However, future management should aim to prevent an increase in inbreeding. Alternatively, even with minimal inbreeding, limited genetic diversity could indicate that the population is at risk due to reduced genetic options.”
28. L393-399: Another contradiction is presented here. One cause of bottleneck events is increased relatedness among individuals, leading to higher homozygosity and inbreeding coefficients. Yet, your results do not reflect this. Please explain this discrepancy.
29. L412: A reference is required for this statement.
In summary, while the revisions have improved the manuscript in several respects, significant work remains, particularly in ensuring accurate data interpretation and proper citation of relevant literature.
Author Response
Thank you for taking the time to review our manuscript. We have uploaded our responses to your questions in the attachment. Please download it at your convenience. Thank you

Reviewer 3 Report (New Reviewer)
Comments and Suggestions for Authors
I have not more comments.
Author Response
Thank you for taking the time to review our manuscript. Thank you.
This manuscript is a resubmission of an earlier submission. The following is a list of the peer review reports and author responses from that submission.
Round 1
Reviewer 1 Report
Comments and Suggestions for Authors
This article focuses on the analysis of genetic diversity and selection signals in Gansu's hu sheep. It evaluates several genetic diversity indicators and employs genome-wide scanning to find candidate genes and SNPs.This article has a competent quantity of effort and a clearer general notion. However, a few matters still need to be settled before publication. Certain methods are stated in an excessively long and repetitive way, we don't need to read a long list of details. Furthermore, there were issues with the English language and several of the results were unclear and poorly interpreted. It is advised that the manuscript be revised and expanded upon.
Comments on the Quality of English LanguageThere were some issues with tenses and grammar, suggested remodify them
Reviewer 2 Report
Comments and Suggestions for Authors
The manuscript analyzed the genetic diversity of Hu sheep breed, basing on different genetic parameters.
It is an interesting study with also breeding and conservation purposes, but unfortunately, the manuscript has several weak points.
First of all, 50 individuals are a very low number to analyze the genetic diversity of a breed. Moreover, how can the calculated parameters be considered accurate and representative if they are based on 50 unrelated samples ? What is the meaning of your F?
For this kind of analysis, the random sampling of a population should be the best approach. Indeed, I think authors are able to describe the diversity of these 50 animals, not of the Hu sheep population.
In the introduction section, remove the part of SNP chips: the chip used here is extremely widespread, so if authors wanted to do a representative list of users, it should be too long.
In addition, remove the subparagraphs in section 2.3. Put all parameters under a single paragraph called e.g. "Genetic diversity analysis", as authors represented results.
You need to justify and improve the section of cluster analysis and pedigree construction. What is the sense using only 50 animals?
Rename CHR in OAR
Are authors sure to give too much importance to MAF distribution? Figure 1 is not necessary, it could be placed in supplementary materials.
Figure 3: to compare results, individuals must be ordered in the same order. Check it.
in PCA, a population stratification is evidenced. Maybe could be necessary to understand which animals are not clustered and why.
Finally, when ROH analysis was performed, a significance threshold must be applied (e.g. top 1% runs, the 70 % of SNPs occurence). Here it is not used and I think is not enough to overlap results to be considered significant.
Comments on the Quality of English LanguageEnglish must be revised. Some terms are not appropriate (e.g. L41 "hybridization", L43 "varieties")
Reviewer 3 Report
Comments and Suggestions for Authors
1. Why did the authors choose 50 rams as experimental samples, given that ewes have a larger population base and greater research value for studying the population?
2. Figure 6A, After analyzing 50 rams using the 50K SNP chip, the Manhattan plot of pi values showed no relevant loci on the Y chromosome. Is this due to a scarcity of SNPs on the Y chromosome within the 50K chip or other reasons?
3. In the absence of phenotype data, relying solely on analysis with the 50K SNP chip to determine associations between certain genes and traits has limited interpretive value. The authors should at least provide frequency information of SNP loci on these selected genes within the 50 rams.
4. Additionally, I encourage the authors to collect phenotype data for correlation analysis in future studies(This is merely a suggestion for future research directions and does not require any modifications to the current manuscript.)